# Effects of Different Factors on the Performance of Recycled Aggregate Permeable Pavement Concrete

**DOI:** 10.3390/ma15134566

**Published:** 2022-06-29

**Authors:** Ruidong Wu, Shuning Shi, Yu Shen, Chen Hu, Mengfei Luo, Zichen Gan, Bin Xiao, Zeyu Wang

**Affiliations:** 1School of Civil and Resource Engineering, University of Science and Technology Beijing, Beijing 100083, China; 18910278944@163.com (S.S.); blessyou0127@163.com (C.H.); invader@163.com (M.L.); rose02110801@163.com (Z.G.); xiaobin1234@163.com (B.X.); wangzeyu0125@163.com (Z.W.); 2Beijing Key Laboratory of Urban Underground Space Engineering, University of Science and Technology Beijing, Beijing 100083, China; 3CCCC-SHEC Third Highway Engineering Co., Ltd., Xi’an 710016, China; shenyu1972@163.com

**Keywords:** recycled aggregate, construction waste, permeability coefficient, pavement concrete, frost resistance

## Abstract

Urban construction has produced a large amount of construction waste which has caused huge environmental problems. The sponge city is the development direction of urban construction, and permeable pavement concrete is an important material for sponge city construction. To see the law influencing different factors on the performance of recycled aggregate permeable pavement concrete, different water binder ratios, recycled aggregate particle gradations, ordinary aggregate substitution rates, and fly ash and admixture contents are designed to prepare permeable concrete. The compressive strength, permeability coefficient, frost resistance, and pore structure of permeable concrete are tested. The results show that when the replacement rate of recycled aggregate is 50%, the 28-d strength of concrete with a 0.25 water binder ratio can reach 28.9 MPa, and the permeability coefficient is 13.26 mm/s. The addition of fly ash will reduce the compressive strength, and the permeability coefficient increases first and then decreases with the increase of the fly ash content. When the mass fraction of fly ash instead of cement is 12%, the 28-d strength is 94.8% of that of the cement group, and the permeability coefficient can reach 14.03 mm/s. A water-reducing agent can obviously improve the workability of permeable concrete; the best content of the water-reducing agent is 0.2% of the cement mass. A reasonable amount of fly ash and water-reducing agent can optimize the number of harmless holes and less harmful holes in the concrete to improve the frost resistance and strength after the freeze–thaw, and the frost resistance is F150. This study provides a theoretical basis and technical guarantee for the resource utilization of recycled aggregate in permeable pavement concrete.

## 1. Introduction

In the process of urbanization, more and more roads are covered by impermeable concrete. The impermeability of these roads has a great impact on the environment, resulting in a series of environmental and safety problems. The groundwater in the area cannot be replenished, resulting in the gradual decline of the groundwater level. When a large amount of rainfall occurs, the surface water cannot be dredged in time, resulting in flood disasters which brings great challenges to the urban drainage system and transportation system. Due to its slow heat dissipation, it is also easy to bring a ‘heat-island effect’ to the city. Building a ‘sponge city’ is the only way in the process of urbanization. Permeable concrete has become one of the most important building materials for the ‘sponge city’ road because of its porosity and good water permeability [1]. At the same time, the infrastructure is constantly updated and iterated, resulting in a large number of solid wastes, such as construction waste. Huge amounts of construction waste are piled up around the city, polluting the environment and causing a waste of resources. The use of construction waste to prepare aggregate for concrete can not only absorb the continuous construction waste but also alleviate the aggregate market gap caused by the prohibition of natural aggregate mining. Recycled aggregate has the characteristics of poor firmness and high-water absorption due to its surface adhesion to cement slurry and rough surface [2], which affects the mechanical strength and durability of concrete [3,4,5]. Permeable concrete has larger porosity and lower requirements for compressive strength than structural concrete. It is completely feasible to use recycled aggregate to prepare permeable pavement concrete [6,7]. 

The researchers have carried out a series of studies on recycled aggregate permeable concrete. The content of recycled aggregate, that is, the substitution rate of recycled aggregate to ordinary aggregate, is the most influential factor on the performance of permeable concrete. Bhutta [8] found that when recycled aggregate is used instead of ordinary aggregate, the permeability coefficient of permeable concrete is much higher, which will also give full play to the performance of permeable concrete. Most scholars’ research shows that recycled aggregate has an adverse impact on the strength of permeable concrete, and the compressive strength decreases with the increase of the proportion of recycled aggregate replacing ordinary aggregate [9,10]. The strength of recycled aggregate is related to the source of construction waste. Recycled aggregate should be used in combination with ordinary aggregate [11,12]. The water permeability and recycled aggregate replacement rate do not show a strong law [13,14]. 

Aggregate gradation, mineral admixture, and the water binder ratio are also important factors affecting the performance of permeable concrete. Adding recycled aggregate with small particle size, the compressive strength of the test block will be relatively high, but its porosity and water permeability coefficient will be relatively low. When the particle size increases gradually, the compressive strength decreases. On the contrary, the porosity and water permeability coefficient will increase. The influence of the gradation of recycled aggregate on the concrete should be fully considered [15]. The permeable concrete prepared using two graded recycled aggregates has good performance [16,17,18]. Güneyisi [19] studied recycled aggregate pervious concrete with water binder ratio of 0.27 and 0.32, respectively. The lower the water binder ratio, the higher the strength of the pervious concrete but the lower the permeability coefficient. Paula [20] prepared the fully recycled aggregate pervious concrete with a water–cement ratio of 0.35, and its performance is good. At present, scholars’ research shows that the water–cement ratio of recycled aggregate permeable concrete ranges from 0.25 to 0.35 [21,22,23]. 

Replacing cement with mineral admixtures, such as fly ash, can improve the performance of recycled aggregate permeable concrete to a certain extent. Peng [24] found that the replacement of cement with fly ash and slag powder will reduce the strength of concrete and reduce the porosity and permeability coefficient. Chang [25] used electric arc furnace slag and alkali-activated slag cement to improve the strength of permeable concrete. Wu [26] believed that fly ash will improve the late strength of pervious concrete and improve the frost resistance of pervious concrete. Zhang [27] found that the proportion of fly ash replacing cement in permeable concrete should not exceed 20%. Liu [28] showed that the compressive strength of fly ash permeable concrete developed well after more than 150 d. Admixtures, such as fly ash [29], pumice powder and nano-clay [30], recycled low-quality brick-concrete [31], pulverized biochar [32], coal ash and rice husk ash [33], or a new admixture [34], were added to prepare environmentally friendly permeable concrete materials. Exceeding the best content will have an adverse impact on the strength and permeability of permeable concrete. According to the current research, there is still a lack of systematic research on the performance of recycled aggregate permeable concrete. Therefore, the systematic study of different influencing factors on the mechanical properties, water permeability, and durability of recycled aggregate permeable concrete is very important for the application of recycled aggregate in pavement permeable concrete.

In this paper, recycled aggregate is used to prepare permeable pavement concrete. The compressive strength, water permeability, and frost resistance of permeable concrete are studied from the aspects of water binder ratio, particle gradation of fully recycled aggregate, replacement rate of recycled aggregate, content of fly ash, and water-reducing agent to find a suitable proportion of recycled bone permeable concrete and optimize its performance indexes. It provides the theoretical basis and technical guarantee for the application of recycled aggregate to prepare pavement permeable concrete in the road construction of a ‘sponge city’.

## 2. Materials and Methods

### 2.1. Raw Materials

Ordinary portland cement is the most commonly used cementitious material for permeable concrete. P.O 42.5 cement produced by the Beijing Jinyu cement plant is selected for this test. All mineral components in cement are shown in Table 1 (the chemical composition of cement is determined by the chemical analysis method of GB/T 176-2008 ‘chemical analysis method of cement’), and the technical indexes of cement are shown in Table 2 (the test method shall refer to GB175-2007 as ‘common portland cement’).

Two kinds of coarse aggregate are used. The recycled aggregate (RA) is selected from the waste concrete of construction waste and processed by crushing and shaping. The ordinary aggregate (OA) is limestone stone with 5–15 mm continuous grading, which meets the national standards (GB/T 14685-2011 ‘pebble and crushed stone for construction’). See Table 3 for specific performance indexes. Table 4 shows the XRF oxide chemical composition of the two aggregates.

The alkali activity of coarse aggregate is an important technical index to measure the aggregate and has an important impact on the durability of concrete. The rapid alkali silica reaction expansion rates of the two aggregates are respectively tested (method in GB/T 14685-2011 ‘pebble and crushed stone for construction’), as shown in Table 5.

According to ‘specification of pebble and crushed stone for building (GB/T14685-2011)’, the 14d expansion rates of the two aggregates shows that they are smaller than 0.10%. Therefore, in most cases, it can be determined that there is no harm of alkali silicate reaction in the two kinds of coarse aggregates.

The grade Ⅰ fly ash produced in Hebei is adopted. Its main chemical composition and related performance data are shown in Table 6 and Table 7 (methods in GB/T 1596-2017 ‘fly ash used for cement and concrete’).

Tap water shall be used as the mixing water and meet the standard requirements. The admixture is the air entraining polycarboxylic acid water reducer produced by Sika Company. The source solution concentration is 40%. Due to the high concentration, it is easy to cause excessive segregation and other problems after adding. It is often diluted to 15% and used in actual construction.

### 2.2. Experiment Methods

The water permeability coefficient is measured with NELD-PC370 equipment. The cube with the size of 100 mm shall be sealed around the permeable concrete with fresh-keeping film. It is necessary to ensure that the top and bottom surfaces of the permeable concrete are used as the permeable surfaces. We install the sealed concrete test block on the permeability coefficient measuring instrument, set the size and water level difference of the permeable concrete test block in the instrument, first start the water pump, and then start the experiment after the water flow is stable. The set time is 1 min. after the water flows through the concrete test block, it flows into the lower bucket, and the permeability coefficient of the concrete test block is obtained by weighing the water quality in the bucket. The standard for determining the permeability coefficient is T/ZACA 001—2018 ‘test method for permeable coefficient of pervious cement concrete’.

The rapid freeze–thaw test shall be carried out in accordance with GB/T 50082-2019 standard. The size of the quick freeze–thaw test piece of concrete is 100 mm × 100 mm × 400 mm. After the concrete is cured for 24 days under standard-curing conditions (temperature 20 ± 2 °C, relative humidity ≥ 95%), soak the concrete in water (temperature 20 ± 2 °C) for four days, and put it into the freeze–thaw machine at 28 days. During freezing and melting, the minimum and maximum temperatures of the center of the test piece shall be controlled at (−18 ± 2) °C and (5 ± 2) °C, respectively. At any time, the center temperature shall not be higher than 7 °C and not lower than −20 °C. After a certain number of freeze–thaw cycles, we test the mass, dynamic elastic modulus, and compressive strength of the specimen. The size of the compressive strength specimen is 100 mm × 100 mm × 100 mm. As the strength grade of permeable concrete is below C30, the loading rate of concrete compressive strength is controlled at 0.3–0.5 MPa/s.

The nuclear magnetic resonance adopts the MesoMR23-060H nuclear magnetic resonance tester with the resonance frequency of 23 MHz. Firstly, the concrete specimen is saturated with water in vacuum for more than 24 h, then the specimen is taken out, and the porosity and pore size distribution of the concrete specimen are tested by nuclear magnetic resonance hydrogen spectrum analysis.

## 3. Results

### 3.1. Effect of Fully Recycled Aggregate Gradation on the Performance of Permeable Pavement Concrete

Two kinds of recycled aggregates with different particle sizes are selected, namely small stone with particle diameter of 5–10 mm and large stone with particle diameter of 10–15 mm. The mass ratios of the two sizes of aggregates are designed to be 2:8, 3:7, 4:6, 5:5, 6:4, 7:3, and 8:2, respectively. The voids are tested with a 10 L capacity cylinder, as shown in Table 8. Then the permeable concrete is prepared with aggregates of different graduation, and the compressive strength and permeability coefficient are tested respectively. The results are shown in Table 9.

The results show that when the ratio of small stone to large stone is 2:8, the porosity is the highest, which is 48.5%. When the ratio of small stone to large stone is 6:4, the porosity is the lowest, which is 41.5%. With the increase of small stone proportion, the porosity value shows a trend of decreasing first and then increasing. The voidage has a great influence on the strength and permeability coefficient of concrete. A more appropriate stone-grading porosity should be selected to ensure good water permeability without affecting the strength.

When the ratio of small stone to large stone is 2:8 and 3:7, the concrete has good water permeability and high-water permeability coefficient, but the strength of the concrete is low. The strength of concrete with 0.25 water binder ratio is the highest when the gradation is 6:4, and the strength is the best when the ratio is 5:5. From the perspective of comprehensive strength and permeability coefficient, it is determined that when the particle gradation of recycled aggregate is 5:5 of small stone and large stone, the performance of concrete is the best. Under the conditions of 0.25 and 0.3 water binder ratio, the 28-d compressive strengths of concrete can reach 27.5 MPa and 21.4 MPa, respectively, and the permeability coefficients are 10.65 mm/s and 12.31 mm/s, respectively.

### 3.2. Effect of Recycled Aggregate Replacement Rate on Permeable Concrete Performance

After the reasonable gradation of recycled aggregate is determined, the content of recycled aggregates in all aggregates are designed to be 0, 30%, 50%, 70%, and 100% respectively. See Table 10 for the mix proportion and results.

It can be seen from Figure 1 that the substitution of recycled aggregate for ordinary aggregate will have an adverse impact on the strength. Recycled aggregate has more defects than natural aggregate; thus, the concrete strength will be reduced [35]. When the replacement rate is 100%, the compressive strength of permeable concrete is the smallest, and the 28-d strengths are 14.4% and 15.7% lower than that of ordinary aggregate permeable concrete, respectively. When the replacement rate is 50% or less, the strength is roughly the same as that of ordinary aggregate; thus, the appropriate replacement rate of recycled aggregate for ordinary aggregate is 50%. The strength of pervious concrete increases with the decrease of the water binder ratio, and the effect of the wwater binder ratio on pervious pavement concrete with different recycled aggregate content is not significant.

Figure 2 shows the variation law of water permeability coefficient with different replacement rates of recycled aggregate. The water permeability coefficient decreases with the increase of the mass fraction of recycled aggregate replacing ordinary aggregate. Recycled aggregate has an adverse impact on the water permeability of permeable concrete. This law is shown for the water binder ratio of 0.3 and 0.25. When the replacement rate of recycled aggregate is 50% or less, it has little effect on the permeability coefficients of permeable concrete, which are only 5.9% and 4.8% lower than that of ordinary aggregate permeable concrete. The reason why the water permeability coefficient of recycled aggregate decreases is that the water absorption of recycled aggregate is much larger than that of ordinary aggregate. Part of the water is absorbed by the aggregate when passing through the concrete, resulting in the decrease of the water permeability coefficient. Combined with the economic and environmental benefits, the reasonable replacement rate of recycled aggregate in permeable pavement concrete is 50%.

### 3.3. Effect of Fly Ash on Permeable Concrete Performance

To study the influence of fly ash content on recycled aggregate permeable pavement concrete, the ratio of water and cementitious material is controlled to be 0.25, the ratio of small stone to large stone in recycled aggregate is 5:5, and the mass ratio of recycled aggregate to ordinary aggregate is 5:5. Different fly ash contents are selected to be 0, 12%, 18%, 24%, and 30% (mass fraction of replacement cement). See Table 11 for mix proportion and test results.

It can be seen from Figure 3 that the addition of fly ash will reduce the compressive strength of permeable concrete, especially the compressive strength of early age, three days and seven days. The activity of fly ash is not as high as that of cement, the early hydration reaction is not as intense as that of cement, and there will be no more hydration products at the early age [36]; thus it will reduce the strength of concrete in early age [37]. When the age reaches 28 days, the compressive strength of fly ash concrete gradually increases, and the strength difference between fly ash concrete and the pure cement group gradually decreases. For example, when the content of fly ash is 12%, the 28-d strength reaches 94.8% of the pure cement group. The permeability coefficient increases first and then decreases with the increase of fly ash content. When fly ash replaces 12% of the cement content, the permeability coefficient is the highest, up to 14.03 mm/s. Therefore, the reasonable content of fly ash in recycled aggregate permeable pavement concrete is 12%.

### 3.4. Effect of Water-Reducing Agent on Permeable Concrete Performance

The designed content of polycarboxylic acid water reducer (PC) is 0–0.5% of the cement mass. See Table 12 for the mix proportion and test results.

The results show that the water-reducing agent can significantly improve the workability of permeable concrete, and the slump of concrete increases significantly with the increase of the water-reducing agent dosage. The appropriate addition of the water reducing-agent can improve the construction performance of recycled aggregate permeable pavement concrete. However, the water-reducing agent will have an adverse impact on the compressive strength of concrete. Taking the 28-day strength as an example, the compressive strength of the control group added with 0.5% water-reducing agent of cement quality is 24.6% lower than that of the control group without water-reducing agent, and the strength decreases significantly. Therefore, the dosage of the water-reducing agent should not be too much. The amount of the water-reducing agent has little effect on the permeability coefficient; this is consistent with reference [38]. Based on the comprehensive analysis of workability, strength, and water permeability, the performance of recycled aggregate permeable pavement concrete is excellent when the dosage of the water-reducing agent is 0.2%.

### 3.5. Frost Resistance of Recycled Aggregate Permeable Concrete

Frost resistance is one of the important durability indexes of permeable concrete. Especially in cold areas, the ability to resist freeze–thaw cycle damage is an important index to ensure the service life of permeable concrete [39,40]. The quality damage and residual compressive strength of pervious concrete after a freeze–thaw cycle are very important for the safe service of concrete [41]. To study the effect of different influencing factors on the frost resistance of recycled aggregate permeable concrete, permeable concrete with recycled aggregate and ordinary aggregate of 0:10, 5:5, and 10:0, concrete with fly ash content of 12%, and admixture of 0.2% are selected to conduct the concrete rapid freeze–thaw test, respectively. The quality, dynamic elastic modulus, and compressive strength of concrete after different freeze–thaw cycles are tested. The results are shown in Table 13 and Table 14.

If the mass loss rate of the concrete is more than 5% of the initial value or the remaining modulus of elasticity of the concrete is less than 60% (in accordance with GB/T 50082-2019 standard). According to the test results, the frost resistances of RA0 and RA50 are F125, the frost resistances of RA100 are F100, and FA12 and PC0.2 are F150. The frost resistance of fully recycled aggregate permeable concrete is the worst, and the frost resistance of 50% substitution rate is the same as that of all ordinary aggregate. Proper amounts of fly ash and the water-reducing agent can improve the frost resistance of permeable pavement concrete, and the loss of quality and dynamic elastic modulus of concrete after a rapid freeze–thaw cycle is the least.

After a certain freeze–thaw cycle, the residual compressive strength of pavement concrete is very important to the bearing capacity of pavement. The compressive strength of concrete after failure of different freeze–thaw cycles is tested, and the results are shown in Table 15.

The residual compressive strength of fully recycled aggregate permeable pavement concrete after different freeze–thaw cycles is the lowest. When freeze–thaw failure occurs, it is only 36.7% of the initial value, which will seriously affect the service life of the pavement. The addition of recycled aggregate will reduce the compressive strength of concrete after a freeze–thaw cycle, but the introduction of fly ash and water reducer will greatly improve the frost resistance and compressive strength of concrete after a freeze–thaw cycle. When the content of fly ash is 12%, the concrete still has 66.8% of the initial strength in case of freeze–thaw failure.

To explore the action mechanism of recycled aggregate, fly ash and water reducer on the frost resistance of permeable pavement concrete, nuclear magnetic resonance (NMR) was used to test the porosity of concrete. The concrete holes are divided according to the hole diameter. Harmless hole is less than 0.02 μm, less harmful is 0.02–0.1 μm, harmful hole is 0.1–0.2 μm, and multi-harmful hole is above 0.2 μm. The porosity calculated according to classification is shown in Table 16.

The freeze–thaw damage of concrete is caused by the volume expansion caused by the ice formed by the water in the internal pores [42,43]. The water in the pores freezes, resulting in the continuous increase of ice pressure and water migration pressure. When this pressure exceeds the tensile strength of concrete, the interior of concrete is damaged. With the continuous development of pore volume expansion, this damage is from the outside to the inside. Pore water freezes in layers, resulting in surface concrete spalling and quality loss. The continuous development of pores leads to the destruction of the internal structure of concrete, which affects the dynamic elastic modulus and compressive strength of concrete.

The frost resistance of fully recycled aggregate permeable pavement concrete is the worst. This is because the number of harmful holes and multi-harmful holes of fully recycled aggregate concrete is significantly higher than that of other concrete. A large number of harmful holes and multi harmful holes accelerate the speed of concrete freeze–thaw damage. Proper addition of fly ash and air-entraining water reducer can increase the content of harmless holes below 0.02 μm; the frost resistance of concrete is directly related to the porosity of harmless holes and multi-harmful holes of concrete before freezing and thawing. That is, the more harmless holes and the fewer multi-harmful holes, the stronger the frost resistance of concrete. The freezing point of capillary water is related to the pore size. The smaller the pore size is, the lower the freezing point of water is. The introduction of harmless holes can play a good buffer role in the freezing and expansion of water in the pores so as to reduce the expansion pressure and slow down the damage of the internal structure of concrete.

## 4. Conclusions

When the proportion of small stone to large stone is 5:5, the performance of concrete is the best. When the water binder ratios are 0.25 and 0.3, the 28-d compressive strengths of concrete can reach 27.5 MPa and 21.4 MPa, respectively, and the water permeability coefficients are 10.65 mm/s and 12.31 mm/s, respectively.The substitution of recycled aggregate for ordinary aggregate will have an adverse impact on the strength. The water permeability coefficient decreases with the increase of the mass fraction of recycled aggregate, replacing ordinary aggregate, and the influence of the water binder ratio on the content of recycled aggregate is not significant. The reasonable mass fraction of recycled aggregate replacing ordinary aggregate is 50%, and the 28-d strength of concrete with a 0.25 water binder ratio can reach 28.9 MPa, and the water permeability coefficient is 13.26 mm/s.The addition of fly ash will reduce the compressive strength, and the permeability coefficient increases first and then decreases with the increase of fly ash content. When the mass fraction of fly ash instead of cement is 12%, the 28-d strength is 94.8% of the cement group, and the permeability coefficient can reach 14.03 mm/s. A water-reducing agent can obviously improve the workability of permeable concrete; the best content of water-reducing agent is 0.2% of the cement mass.The frost resistance of fully recycled aggregate concrete is the worst, only reaching F100. Adding an appropriate amount of fly ash and air-entraining water reducer can greatly improve the frost resistance and freeze–thaw strength of concrete, and the frost resistance can reach F150. The strengthening mechanism is to effectively increase the number of harmless holes and less harmful holes in the concrete and optimize the internal hole structure of the concrete.

This study reveals the action laws of different influencing factors on the compressive strength, water permeability, frost resistance, and pore structure of recycled aggregate permeable concrete and provides a theoretical basis for the engineering application of recycled aggregate in permeable pavement concrete materials.

## Figures and Tables

**Figure 1 materials-15-04566-f001:**
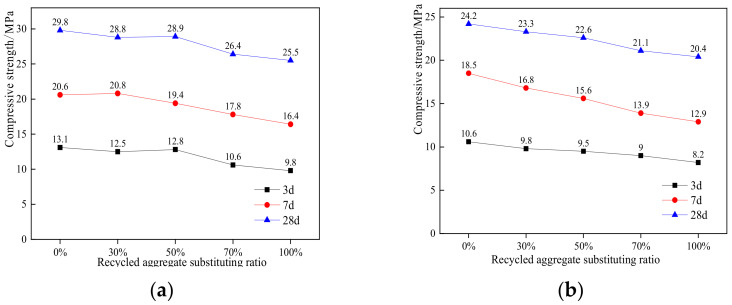
Strength of permeable concrete with different replacement rates of recycled aggregate: (**a**) 0.25 w/c, (**b**) 0.3 w/c.

**Figure 2 materials-15-04566-f002:**
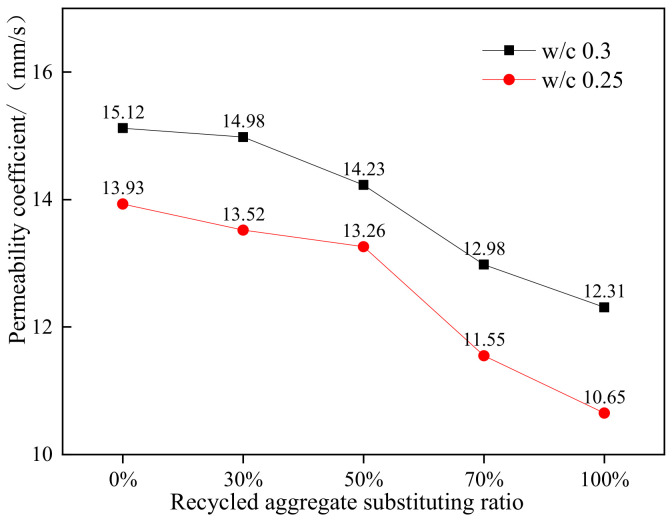
Water permeability coefficient of concrete with different replacement rates of recycled aggregate.

**Figure 3 materials-15-04566-f003:**
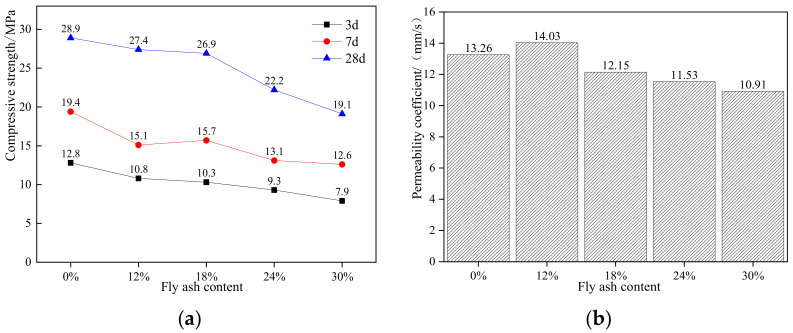
Strength and water permeability coefficient of concrete with different fly ash content: (**a**) strength, (**b**) water permeability coefficient.

**Table 1 materials-15-04566-t001:** Mineral composition of cement/%.

Mineral Composition	C_3_S	C_2_S	C_3_A	C_4_AF	f-CaO	f-MgO
Content	61.1	18.3	7.2	8.7	0.9	1.7

Note: The content of gypsum in cement is 2.95–3.0%.

**Table 2 materials-15-04566-t002:** Main performance indexes of cement.

Project	Density /(g/cm^3^)	Loss on Ignition/%	Setting Time/min		Soundness (Ray)	Compressive Strength/MPa		Flexural Strength/MPa	
			Initial	Final		3 d	28 d	3 d	28 d
P·O 42.5	3.06	3.78	145	210	Qualified	32.2	53.1	5.6	7.8

**Table 3 materials-15-04566-t003:** Performance indexes of recycled aggregate and ordinary aggregate.

Sample Name	Bulk Density/(g/cm^3^)	Apparent Density/(g/cm^3^)	Porosity /%	Sediment Percentage/%	Needle Flake Particle Content/%	Crushing Value/%	Water Absorption /%
RA	1.44	2.57	42.6	2.5	3.9	11.6	7.1
OA	1.46	2.45	40.3	2.3	2.9	8.7	3.6

**Table 4 materials-15-04566-t004:** XRF chemical composition analysis of two aggregates/%.

Chemical Composition	CaO	MgO	Al_2_O_3_	SiO_2_	P_2_O_5_	SO_3_	Fe_2_O_3_	K_2_O
RA	85.47	0.83	3.77	9.02	0.34	0.13	2.27	0.97
OA	55.21	0.71	12.65	17.62	0.21	0.17	1.38	0.68

**Table 5 materials-15-04566-t005:** Rapid alkali silica reaction expansion rates of two aggregates.

Sample	14 d Expansion Rate	Result Judgment
RA	0.005%	No potential alkali silica reaction hazard
OA	0.006%	No potential alkali silica reaction hazard

**Table 6 materials-15-04566-t006:** XRF main chemical composition of fly ash/%.

Composition	Al_2_O_3_	MgO	SiO_2_	Fe_2_O_3_	K_2_O + Na_2_O	CaO	Other
Mass fraction	25.8	1.2	52.7	9.7	2.3	3.7	4.6

**Table 7 materials-15-04566-t007:** Main performance indexes of fly ash.

Sample	Water Demand Ratio/%	Density/(g/cm^3^)	Fineness (45 μm Sieve Residue)/%	Specific Surface Area/(m^2^/kg)	Loss on Ignition/%
Fly ash	98	2.23	8.7	441	2.2

**Table 8 materials-15-04566-t008:** Porosity of different particle size gradation of fully recycled aggregate.

Small Stone:Large Stone	Small Stones Mass/kg	Large Stones Mass/kg	Volume of Water/L	Voidage/%
2:8	2.25	8.80	4.85	48.5
3:7	3.50	8.02	4.55	45.5
4:6	4.60	6.95	4.40	44.0
5:5	5.82	5.79	4.25	42.5
6:4	7.05	4.68	4.15	41.5
7:3	8.06	3.45	4.45	44.5
8:2	9.20	2.35	4.50	45.0

**Table 9 materials-15-04566-t009:** Performance of pervious concrete with different fully recycled aggregate gradation.

Small Stone:Large Stone	Cement /(kg/m^3^)	Total Aggregate /(kg/m^3^)	Water /(kg/m^3^)	Compressive Strength/MPa			Porosity /%	Permeability Coefficient /(mm/s)
				3 d	7 d	28 d		
2:8	400	1400	100	8.7	17.8	25.5	11	10.56
			120	8.1	15.5	20.8	14	15.53
3:7	400	1400	100	9.2	16.9	24.4	12	12.11
			120	8.3	15.4	20.6	14	14.53
4:6	400	1400	100	9.6	18.6	26.7	10	10.29
			120	8.2	16.8	21.1	13	13.96
**5:5**	**400**	**1400**	**100**	**10.5**	**19.4**	**27.5**	**11**	**10.65**
			**120**	**8.7**	**16.9**	**21.4**	**12**	**12.31**
6:4	400	1400	100	10.2	20.1	28.6	9	8.68
			120	8.4	16.4	20.6	10	10.32
7:3	400	1400	100	8.3	18.8	24.9	11	11.15
			120	7.6	15.6	20.7	13	13.08
8:2	400	1400	100	7.3	16.5	23.9	12	11.55

**Table 10 materials-15-04566-t010:** Concrete mix proportion with different recycled aggregate substitution rates.

Group	Cement /(kg/m^3^)	OA /(kg/m^3^)	RA /(kg/m^3^)	Water /(kg/m^3^)	Compressive Strength/MPa			Porosity /%	Permeability Coefficient /(mm/s)
					3 d	7 d	28 d		
RA0	**400**	1400	0	100	13.1	20.6	29.8	13	13.93
				120	10.6	18.5	24.2	14	15.12
RA30		980	420	100	12.5	20.8	28.8	13	13.52
				120	9.8	16.8	23.3	14	14.98
**RA50**		**700**	**700**	**100**	**12.8**	**19.4**	**28.9**	**13**	**13.26**
				**120**	**9.5**	**15.6**	**22.6**	**14**	**14.23**
RA70		420	980	100	10.6	17.8	26.4	11	11.55
				120	9.0	13.9	21.1	13	12.98
RA100		0	1400	100	9.8	16.4	25.5	11	10.65
				120	8.2	12.9	20.4	12	12.31

**Table 11 materials-15-04566-t011:** Mix proportion and performance of permeable concrete with different fly ash content.

Group	Cement /(kg/m^3^)	Fly Ash /(kg/m^3)^	Aggregate /(kg/m^3^)	Water /(kg/m^3^)	Compressive Strength/MPa			Porosity /%	Permeability Coefficient /(mm/s)
					3 d	7 d	28 d		
FA0	400	0	1400	100	12.8	19.4	28.9	13	13.26
**FA12**	**352**	**48**	**1400**	**100**	**10.8**	**15.1**	**27.4**	**13**	**14.03**
FA18	328	72	1400	100	10.3	15.7	26.9	12	12.15
FA24	304	96	1400	100	9.3	13.1	22.2	12	11.53
FA30	280	120	1400	100	7.9	12.6	19.1	11	10.91

**Table 12 materials-15-04566-t012:** Mix proportion and performance of permeable concrete with different admixture content.

Group	PC Content/%	Cement /(kg/m^3^)	Aggregate /(kg/m^3^)	Water /(kg/m^3^)	Compressive Strength/MPa			Porosity /%	Permeability Coefficient /(mm/s)	Slump/mm
					3 d	7 d	28 d			
PC0	0	400	1400	100	12.8	19.4	28.9	13	13.26	30
PC0.1	0.1	400	1400	100	11.3	16.9	27.2	13	13.75	45
PC0.2	0.2	400	1400	100	11.6	15.8	27.6	13	13.88	55
PC0.3	0.3	400	1400	100	11.8	15.4	24.1	13	13.83	70
PC0.4	0.4	400	1400	100	10.8	14.7	22.5	14	13.82	95
PC0.5	0.5	400	1400	100	10.3	14.1	21.8	14	13.93	110

Note: The PC content is the mass ratio of cement.

**Table 13 materials-15-04566-t013:** Mass loss rate of concrete after different freeze–thaw cycles/%.

Number of Cycles	25	50	75	100	125	150	175
RA0	0.2	0.6	1.0	1.3	3.8	4.7	
RA50	0.4	0.8	1.9	2.7	4.6	**5.9**	
RA100	1.2	1.9	2.8	4.1	**5.4**		
FA12	0.2	0.4	0.7	1.1	1.6	3.5	4.8
PC0.2	0.2	0.5	0.8	1.3	2.8	4.1	**5.2**

Note: Bold font is after freeze–thaw damage has occurred.

**Table 14 materials-15-04566-t014:** Residual relative dynamic elastic modulus of concrete after different freeze–thaw cycles/%.

Number of Cycles	25	50	75	100	125	150	175
RA0	95.2	90.1	83.1	77.6	70.5	**58.1**	
RA50	93.8	87.2	79.5	73.1	66.5	**55.6**	
RA100	88.1	75.9	64.2	46.5	**55.1**		
FA12	98.1	92.5	88.7	79.6	72.2	66.4	**56.3**
PC0.2	96.6	91.6	85.6	73.5	68.1	62.2	**54.8**

Note: Bold font is after freeze–thaw damage has occurred.

**Table 15 materials-15-04566-t015:** Compressive strength of concrete after different freeze–thaw cycles/MPa.

Number of Cycles	0	25	50	75	100	125	150	175
RA0	29.8	27.2	23.4	20.1	19.6	18.9	15.1	
RA50	27.9	24.8	22.2	19.5	16.1	15.5	11.6	
RA100	27.5	22.5	18.9	15.2	13.5	10.1		
FA12	27.4	26.2	24.2	23.7	21.3	20.7	19.5	18.3
PC0.2	27.6	26.6	22.0	21.1	20.5	18.9	18.1	16.8

**Table 16 materials-15-04566-t016:** Hole structure analysis of concrete/%.

	Total Porosity	<0.02 μm	0.02–0.1 μm	0.1–0.2 μm	>0.2 μm
RA0	12.47	4.68	1.29	0.56	5.94
RA50	13.73	4.24	1.13	1.21	7.15
RA100	15.66	3.11	0.76	4.63	6.96
FA18	11.12	6.88	1.21	0.73	2.30
PC0.2	11.61	5.79	0.84	1.12	3.86

## Data Availability

The original contributions presented in the study are included in the article; further inquiries can be directed to the corresponding author.

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
