# Peer review of "Effects of Different Factors on the Performance of Recycled Aggregate Permeable Pavement Concrete"

_materials, 2022, doi:10.3390/ma15134566_

Round 1

Reviewer 1 Report

Materials-1755243:

*The title is too broad. Effects on which aspects?

*The abstract must be rewritten, as it does not contain all elements. A good abstract should have: a brief introduction (research justification); main goal; global method; main results; and, general conclusion.

*Keywords are suitable.

*The initial part of the introduction is didactic and not scientific.

*A good introduction should show the state of the art; what has been done so far on the topic; what still needs to be investigated for the advancement of knowledge; what is the existing gap? The introduction does not bring this approach and must be rewritten.

*What is the originality of this paper?

*2. Materials and Methods:

-Justify the choice of cement?

-What technique was used to determine the composition of table 1?

-Present the standards for the tests in tables 2 and 3.

-Table 4 presents the composition in terms of oxides, make this clear in the text.

-What was the technique used to determine the properties shown in table 5?

-Table 5: better present the graph.

-What are the techniques used to determine the chemical composition of table 6 and the physical characteristics of table 7?

-Line 120: Present information about the polycarboxylic. Why 15%? Justify.

*2.2 Experiment methods:

-Figure 1 can be deleted.

-Lines 122-132: What is the standard for determining the permeability coefficient?

-Does the experimental program only consist of 2 tests? Permeability and freeze-thaw? With these two tests it is possible to reach what scientific contribution to the area?

*3. Results:

-Lines 156-162: The discussion is shallow and unscientific. Improve.

-In table 9 the authors present results without standard deviation or error.

-Lines 164-172: The discussion of the results does not consider the error. It lacks depth.

-Table 10: Same as the comment made for table 9.

-Lines 182-206: Authors should compare the results with the existing literature.

-Table 11 to 16: Same as the comment made for tables 9 e 10.

-Lines 219-230: Authors should compare the results with the existing literature.

-In general, the authors must re-discuss the results, compare them with the literature, and explain the findings on a scientific basis. So far, the discussion is similar to a technical report.

*4. Conclusions: The authors replicate the discussion of the results. A good conclusion is one that answers the general objective of the research concisely.

Author Response

Dear Reviewer,

Thank you for your comments and suggestions. Your comments are of great help to the improvement of our manuscript. The authors have carefully discussed your comments and revised the manuscript according to your requirements, using the revision mode in the attachment. The following response is made to the question. Please see the attachment.

Reviewer 2 Report

Dear Authors 

Your study is rather simple but it has both: scientific and educational value.

I do not have any major concerns about the value and presentation of undertaken experiments. I was pretty surprised that you do not refer to your previous studies as if it was your first time with the recycled concrete issue.

The list of my general comments and some minor editorial concerns are given below.

1. Introduction

Please try to avoid "cluster citations" like [9-12], [19-23] and [24-28]. Every referenced paper deserves to be properly introduced to the Reader. Personally, I'd add a bunch of general comments on reuse of crushed concrete, starting from its direct application in geotechnical engineering to form embankments on the building site (which is the least absorbing way of using it, concerning lack of transport and storing cost). Lots of valuable contributions were written by C.S. Vieira from Portugal, S. Kwiecien from Poland and other Authors (easy to find in Scopus). It is not mandatory, but it could attract wider attention to your study.

2. Materials and Methods

This section is cautiously written, concerning material characteristics.

I'd expect a more precise description of strength testing methodology. In lines 142-144 we can read: "After a certain number of freeze-thaw cycles, test the mass, dynamic elastic modulus and compressive strength of the specimen. The size of the compressive strength specimen is 100 mm × 100 mm × 100 mm." This information seems to be incomplete. I could not find any "elastic modulus results" in following sections of your work.

Please provide information about speed of loading (if possible). An exemplary strain-stress graph would be appreciated, with methodology of its analysis concerning elastic modulus and strength of material.

3. Results

Please use a coherent description of units in all Tables.

I observe that you briefly follow the IMRaD structure of scientific contribution. But your study is missing a real Discussion of your results in the light of other Researchers' findings (introduced and referenced in the introductory part). That decreases the clarity of your presentation and a value of your findings, as they are not properly confronted with current "State of the Art" in the discipline.

I have some concerns about the final time of your testing. I understand that 3, 7 and 28 days are standard times in most of codes describing concrete testing worldwide. From my experience, non-standard cement/concrete composites achieve their final capacity and stiffness after 2-3 months. I understand that nowadays, you will not be able to add this information, but you might at least make some reservations concerning the "time issue" and maybe show some prospects for further research.

4. Conclusions

I observe again that your conclusions are proper but a little bit trivial and simply confirming common sense and basic engineering judgement. That is not an objection but just an incentive to continue more profound studies with regard to future practical outcomes.

5. Reference list

The choice of the references is always Authors' right and responsibility. I was pretty surprised that I could not find any self-citations as if Authors never performed similar studies before, But this is not a problem. 

I noticed that references to Chinese authors form 66% of the reference list. Most of the references seem to be pretty relevant but I'd strongly recommend to make a short survey in Scopus, looking for more international and recent references. It always widens the group of potential Readers and, last but not least, raises the citing potential of your study.

Best regards

I marked "major revision" just because of the number of my comments. 

Best regards

Author Response

Thank you for your comments and suggestions. Your comments are of great help to the improvement of our manuscript. The authors have carefully discussed your comments and revised the manuscript according to your requirements, using the revision mode in the attachment. The following response is made to the question. Please see the attachment.

Round 2

Reviewer 1 Report

The authors did a good review job.

Reviewer 2 Report

Dear Authors

I cannot find any reason to object publication of the revised version of your manuscript. From my perspective, you introduced necessary corrections and/or provided reasonable explanation why you did not follow my recommendations. Variety of "view points" is a value in science :-)

I opt for acceptance in present form

Best regards